# The Pharmacokinetics in Mice and Cell Uptake of Thymus Immunosuppressive Pentapeptide Using LC-MS/MS Analysis

**DOI:** 10.3390/molecules27134256

**Published:** 2022-07-01

**Authors:** Shang Chen, Chenyan Ren, Yuan Ji, Dongke Liu, Xinke Zhang, Fengshan Wang

**Affiliations:** 1Key Laboratory of Chemical Biology (Ministry of Education), NMPA Key Laboratory for Quality Research and Evaluation of Carbohydrate-based Medicine, Institute of Biochemical and Biotechnological Drug, School of Pharmaceutical Sciences, Cheeloo College of Medicine, Shandong University, Jinan 250012, China; cswarm@163.com (S.C.); 15966331522@163.com (C.R.); liudongke951107@163.com (D.L.); 2Shandong Institute for Food and Drug Control, Jinan 250101, China; yuanji91@outlook.com; 3Key Laboratory of Chemical Biology (Ministry of Education), Department of Pharmacology, School of Pharmaceutical Sciences, Cheeloo College of Medicine, Shandong University, Jinan 250012, China; 4NMPA Key Laboratory for Quality Research and Evaluation of Carbohydrate-based Medicine, National Glycoengineering Research Center, Shandong University, Jinan 250012, China

**Keywords:** LC-MS/MS, bioanalytical method development, allergic asthma, pharmacokinetics, cell uptake

## Abstract

Thymus immunosuppressive pentapeptide (TIPP) is a novel anti-inflammatory peptide with high efficacy and low toxicity. This study aims to establish a selective LC-MS/MS method for analyzing the analyte TIPP in biological samples, laying the foundation for further PK and PD studies of TIPP. Protein precipitation was conducted in acetonitrile supplemented with 2% formic acid and 25 mg/mL dithiothreitol as a stabilizer, which was followed by backwashing the organic phase using dichloromethane. The chromatographic separation of TIPP was achieved on a C18 column with a gradient elution method. During positive electrospray ionization, TIPP was analyzed via multiple-reaction monitoring. The linear relationships between the concentration of TIPP and peak area in murine plasma cell lysates, supernatants, and the final cell rinse PBS were established within the ranges of 20–5000 ng/mL, 1–200 ng/mL, 10–200 μg/mL, and 0.1–20 ng/mL, respectively (*r*^2^ > 0.99). Validated according to U.S. FDA guidelines, the proposed method was proved to be acceptable. Such a method had been successfully applied to investigate the pharmacokinetics of TIPP in mice via subcutaneous injection. The plasma half-life in mice was 5.987 ± 1.824 min, suggesting that TIPP is swiftly eliminated in vivo. The amount of TIPP uptake by RBL-2H3 cells was determined using this method, which was also visually verified by confocal. Furthermore, the effective intracellular concentration of TIPP was deduced by comparing the intracellular concentration of TIPP and degrees of inflammation, enlightening further investigation on the intracellular target and mechanism of TIPP.

## 1. Instruction

Allergic asthma is a prevalent disease associated with chronic airway inflammation [1,2]. Asthma is intractable but manageable with drugs. Currently, inhaling corticosteroids is commonly used to treat asthma, but its clinical application is restricted by adverse effects [1,2,3,4]. Hence, it is urgent to develop novel drugs of high efficacy and low toxicity for asthma. Previously, our group found a new peptide, thymus immunosuppressive pentapeptide (TIPP, shown in Figure 1a), from calf thymus extracts, which exhibited anti-inflammatory effects both in vitro and in vivo. Of relevance to these studies, immunoglobulin E (IgE)-mediated activation of rat basophilic leukemia cells (RBL-2H3) was suppressed by TIPP, suggesting that TIPP has a protective anti-allergic effect in vitro [5]. Furthermore, in a murine model of ovalbumin (OVA)-induced allergic asthma, TIPP (50 mg/kg) showed the curative effect on airway inflammation with a survival rate of 100% [6]. It has been reported that small peptides mostly take effect by binding to receptors on the cell membrane rather than entering cells [7]. However, according to our previous research, TIPP was likely to exert its pharmacological effects by binding to proteins within cells. Therefore, the cell entry of TIPP needs to be verified, which will pave the way for the target identification and mechanistic research of TIPP.

The pharmacokinetic properties in animals and its action mechanism also need to be studied using reliable, accurate, and sensitive methods in order to develop TIPP into a new drug. Short peptides can be analyzed using fluorescence labeling, immunoassay, and liquid chromatography-tandem mass spectrometry (LC-MS/MS) [8,9,10]. Although the visualization and quantification of fluorescently labeled peptides are widely used, there are still some drawbacks. One such flaw is that fluorescent labeling might change the structure of small molecules, resulting in the alteration of physical and chemical properties as well as activity. The performance of immunoassay strongly relies on the specificity of antibodies, which is too complicated, onerous, and time-consuming for a new small peptide. In contrast, the combination of LC and MS/MS based on the excellent chromatographic separation power and highly selective detection ability is competent for sensitive and high-throughput analyses. However, there are several challenges for the analysis of peptides in biological samples using LC-MS/MS, including sample complexity and peptide stability. Relative quantification strategies are employed in most quantitative studies of peptides. Chemical peptide analogs or stably isotope-label peptide analogs are generally used as the internal standards for quantitative multiple reaction monitoring (MRM) analysis. Previous results suggested that the stability of TIPP would be significantly improved when the -SH group was substituted with the -OH group, namely S4 (Figure 1b) [11]. So, S4, the analog of TIPP, was used as the internal standard (IS) of this method.

The study aimed to develop a highly sensitive and selective LC-MS/MS method for the quantitative analysis of TIPP in murine plasma, RBL-2H3 cell lysates, cell supernatants and the final cell rinse PBS, so as to be used to study the preclinical pharmacokinetics (PK) in mice and cell uptake of TIPP. To investigate the dynamic characteristic of TIPP in cells, the concentration of TIPP in cell lysates was determined. In order to guarantee the accuracy, the cells were washed five times, which could eliminate the influence of extracellular TIPP or adhered TIPP on the cell membrane. Thus, TIPP in the final cell rinse was quantitated in order to guarantee the extracellular TIPP was cleaned up. Fluorescein isothiocyanate (FITC)-labeled TIPP was visualized using a confocal microscope to verify TIPP uptake in cells. Combined with timing pharmacodynamics (PD) index studies, such as degranulation marker β-hexosaminidase, interleukin-4 (IL-4), and histamine [12,13], the effective intracellular concentration in RBL-2H3 cells was determined. This study is the first report of a quantitative method for the bioanalysis of TIPP, laying a foundation for further physiological and pharmacological studies as well as providing novel strategies for quantitative analysis of the effective concentration of drugs in vitro.

## 2. Results

### 2.1. Optimization of LC-MS/MS Conditions

In order to obtain better separation efficiency and maximal peak resolution, various chromatographic conditions were investigated including mobile phase (Acetonitrile (ACN), methanol (MeOH), and three times distilled water (TDW)), additives (formic acid (FA), acetic acid, and trifluoroacetic acid), and gradient parameters. The ionic strength and sensitivity of TIPP in mass spectra could be enhanced with the addition of FA. Therefore, 0.1% FA-TDW and 0.1% FA-ACN were selected as the mobile phase. Through gradient elution on the Thermo Hypersil Gold C18 column (2.1 mm × 100 mm, 5 μm), TIPP and S4 were separated with good peak shapes (Figure 2a). Furthermore, high-intensity analyte peaks, good reproducibility, and high resolution were also achieved through gradient elution. Therefore, chromatographic separation was conducted with a mobile phase consisting of 0.1% FA-ACN (10–90%) and 0.1% FA-TDW in a gradient mode (Table 1).

Under acidic conditions, TIPP and S4 were more prone to protonation than deprotonation in MS. Therefore, the positive mode was chosen. The main molecular ions of TIPP and S4 were located at *m*/*z* 605.3 and *m*/*z* 589.3, respectively (Figure 2b,d). These protonated molecular ions were further fragmented by nitrogen in the collision cell of the triple quadrupole mass spectrometer, and the main product ions were at *m*/*z* 490.2 for TIPP and *m*/*z* 474.3 for S4, respectively (Figure 2c,e). Hence, *m*/*z* 605.3 > 490.2 for TIPP and *m*/*z* 589.3 > 474.3 for S4 were used for TIPP quantification in the MRM mode.

### 2.2. Optimization of Sample Preparation

For protein precipitation in murine plasma, 2% FA-ACN had better performance than MeOH. In order to achieve better precipitation, the proportions of precipitant added to plasma was screened. Although the intensity of the TIPP ion peak was negatively correlated with the volume of 2% FA-ACN in aqueous solution, it showed the opposite trend in plasma samples, implying that a large volume of 2% FA-ACN achieves higher efficiency in protein precipitation in plasma. The results of protein precipitation with only organic reagents showed that TIPP remained stable in aqueous solution and maintained a high response in 10 min, whereas the intensity decreased significantly in plasma under the same conditions. The results of backwashing of highly water-soluble TIPP with dichloromethane (DCM) significantly increased the intensity of TIPP. Therefore, 300 μL of 2% FA-ACN combined with DCM extraction was determined as the optimal pretreatment method (Figure 3a).

Recovery of TIPP was evaluated in samples with different stabilizers, including DTT, L-AA, and PMSF. The results showed that DTT had a better stabilizing effect than other stabilizers (Figure 3b). The concentration of DTT was further optimized, and 20 mg/mL DTT was finally used as the optimum stabilizer, for its average recovery was higher than 95% (Figure 3c).

### 2.3. Methodology Validation

The performance of the proposed HPLC-MS/MS method was assessed on its selectivity, linearity, accuracy, precision, recovery, matrix effect, and stability.

#### 2.3.1. Selectivity

No interference peak in the channels of *m*/*z* 605.3 > 490.2 and *m*/*z* 589.3 > 474.3 was observed in blank samples of four different matrices. In the lowest limit of quantification (LLOQ) samples, no significant interference or co-elution peak exceeding 20% of the analyte area, or co-elution peak exceeding 5% of the IS area was observed, suggesting the proposed method had good selectivity.

#### 2.3.2. Linearity

The representative standard curves of TIPP quantification in plasma samples, cell lysates, cell culture supernatants, and the final cell rinse are shown in Appendix A. The *r*^2^ of the linear equations of the standard curve in different samples was all greater than 0.99, indicating good linearity, and the linear range met the requirements of PK and PD analysis in vivo and in vitro.

#### 2.3.3. Accuracy and Precision

The accuracy and precision of TIPP determination in different matrices samples were analyzed based on QC data at four different concentrations (Table 2). The intra-assay precision and inter-assay precision in low-quality control (LQC), medium-quality control (MQC), and high-quality control (HQC) samples and accuracy at each level were within ±15%, with LLOQ lower than ±20%, which met the requirements. The inter-assay and inter-assay accuracy (%Bias) in different matrix samples were −8.22–3.02% and −5.28–4.90%, respectively. The inter-assay and inter-assay precisions (%RSD) were 1.56–10.62% and 2.75–7.91%, respectively.

#### 2.3.4. Matrix Effect and Recovery

The matrix effect and recovery performed in murine plasma, cell lysates, and cell culture supernatants at three QC levels are summarized in Table 3. The mean IS-normalized matrix factor (MF) in different samples were found to be in the range of 97.9–113.7%, which were reproducible within the acceptable %CV limits of 15%. Additionally, TIPP and IS in the same samples showed similar extraction recovery, featuring ranges of 41.1–42.6% in murine plasma, 80.2–94.1% in cell culture supernatants, and 57.4–68.2% in cell lysates.

#### 2.3.5. Stability

The stability of TIPP in processed murine plasma samples was studied under four different conditions (Table 4). The results showed that TIPP remained stable at room temperature for 4 h, in an autosampler (at 4 °C for 24 h), three freeze–thaw cycles and at −80 °C for four weeks.

### 2.4. PK of TIPP via Subcutaneous Injection in Mice

The validated method was successfully applied to the preliminary PK study of TIPP in mice. At around 40 min after subcutaneous injection, the concentration of TIPP was below the limit of detection. The mean plasma concentration-time curve was shown in Figure 4. The AUC, elimination half-life (*t*_1/2_), peak concentration (*T*_max_), maximum plasma concentration (*C*_max_), mean residence time (MRT), apparent volume of distribution (*V*_d_/F), and apparent clearance rate (CL/F) were calculated using STATA software (Table 5). *C*_max_ (623 ng/mL) was observed at 10 min (*T*_max_), demonstrating that TIPP could be quickly absorbed after subcutaneous injection. The mean *t*_1/2_ of TIPP was 5.987 min representing TIPP could be eliminated quickly.

### 2.5. Uptake and Activity of TIPP in RBL-2H3 Cells

#### 2.5.1. Uptake of TIPP in RBL-2H3 Cells

According to the drug versus time curve of TIPP in RBL-2H3 cells (Figure 5a), when incubating with 200 μg/mL TIPP, the concentration of TIPP in cell culture supernatant gradually declined and was below the detection limit at 12 h. The intracellular concentration of TIPP increased to 24.57 ± 4.53 ng/mL at 20 min and achieved a plateau (38.62 ± 2.47 ng/mL) from 30 min onwards (Figure 5b). No residual TIPP was detected in the final cell rinse PBS, indicating that there was no residual TIPP in the supernatants to interfere with the detection of TIPP in the cell lysates. In order to further explore intracellular TIPP degradation, the TIPP-free medium was substituted after the intracellular concentration of TIPP reached a plateau. Then, the cell lysates were harvested at different time points to examine the concentration of TIPP. In addition, the RBL-2H3 cells were rinsed, which could prevent extracellular TIPP interference. The intracellular TIPP concentration declined by over 50% within 5 min (*t*_1/2_ = 5.596 min), and the compound was almost eliminated within 30 min (Figure 5c).

The confocal pictures of RBL-2H3 cells treated with FITC-TIPP were in accordance with the TIPP uptake study by LC-MS/MS. FITC-TIPP with the same molar concentration as 200 μg/mL TIPP could be co-localized with RBL-2H3 cells at 10 min, and the fluorescent signal remained steady after 30 min with significant intranuclear enrichment (Figure 5d).

#### 2.5.2. In Vitro Activity of TIPP

Extracellular β-hexosaminidase, histamine, and IL-4 can be used as biomarkers of degranulation and inflammation. As illustrated in Figure 6, compared to unstimulated cells, the IgE–antigen complex significantly increased the levels of β-hexosaminidase, histamine, and IL-4 in RBL-2H3 cells. The allergic response was also increased with exposure duration. Pretreatment with 200 μg/mL TIPP for 20 min significantly decreased the level of β-hexosaminidase, histamine, and IL-4, which demonstrated that TIPP could inhibit the degranulation and inflammation induced by IgE–antigen complex in vitro.

## 3. Discussion

TIPP is a novel compound for treating asthma discovered by our group. Yet the analysis of TIPP in biological matrix samples has not been investigated. In this study, an LC-MS/MS-based method for the quantitative analysis of TIPP in murine plasma and RBL-2H3 cells was developed and validated, laying the foundation for further PK and PD studies of TIPP.

In this study, the optimal pre-processing method and analysis conditions were optimized. Due to complex matrices in blood and cell samples, the efficient separation of target peptide and IS without affecting their stability is essential for pre-processing [14]. As a common protein precipitation agent, ACN had better precipitation efficiency when the volume ratio of ACN to the matrix samples was 3:1. The addition of 2% FA further improved the peak shape and increased the ionization efficiency of TIPP. However, TIPP is a small peptide with limited solubility in the organic phase, and thus, DCM was supplemented to remove the organic phase [15,16]. Moreover, due to the instability observed between batches, the performance of stabilizers was evaluated, among which DTT had a better stabilizing effect. The sulfhydryl group as the active site in TIPP is prone to cross-linking in complex biological matrix samples, making it less likely to be detected by MS. In contrast, as a reducing agent, DTT can effectively prevent the occurrence of cross-linking [10,17]. Compared with the commonly used SPE columns in the processing of water-soluble protein samples [18,19], this method is more economical and environmentally friendly, by which higher intensity and better specificity of TIPP can also be obtained. In addition, the composition of the mobile phase and chromatographic column were also optimized to achieve high sensitivity and selectivity in LC-MS/MS. The mobile phase was supplemented with 0.1% FA to enhance the intensity and selectivity of TIPP in LC-MS/MS. The ion pairs of the selected target and IS had a stronger response after optimizing indicators, including collision energy and declustering potential in the MRM mode [10,20,21]. This enabled the biological analysis of TIPP with high precision and sensitivity, and it developed new ideas for the quantitative analysis of small peptide compounds with high water solubility. The results of methodology validation also showed that the developed method was competent for analyzing TIPP in murine plasma and RBL-2H3 cells, which is of great importance for further PK-PD study.

This method had been successfully applied to study the PK of subcutaneously administered TIPP in mice. PK results revealed that *T*_max_ in murine plasma was 10 min, implying that TIPP could be rapidly absorbed into the blood and reach the *C*_max_ of 623 μg/L. After 40 min, TIPP concentration decreased to about 20 ng/mL. The half-life of TIPP was (5.987 ± 1.824) min. The swift absorption and elimination of TIPP indicated that TIPP might serve as a hormone in mice. TIPP takes effect and degrades rapidly after injection, making it appealing as a potential drug with lower side effects to treat chronic inflammatory diseases such as asthma.

To confirm whether TIPP takes effect intracellularly, an analysis of TIPP was carried out with RBL-2H3 cells. The study was designed based on the well-developed quantitative method for TIPP in murine plasma, so the methodology validation was further optimized in vitro. Eventually, a reliable method for the quantitative detection of TIPP in RBL-2H3 cells was obtained. The result showed that the concentration of TIPP gradually reduced in cell culture supernatants, suggesting that TIPP is unstable in the extracellular matrix, which is consistent with the short *t*_1/2_ observed in animal experiments. Moreover, the concentration of TIPP was below the detection limit of 0.1 ng/mL in all PBS rinse samples, indicating that there was no TIPP residue in the supernatants. The analysis of cell lysates proved that TIPP indeed bounded to or entered RBL-2H3 cells. In addition, the cells were first treated with TIPP for 40 min, and the intracellular concentration of TIPP decreased to near LLOQ about 30 min after removing TIPP in the cell medium, which suggests that TIPP also degrades rapidly in the cells. Confocal images confirmed the cell entry of labeled TIPP as well. These lines of evidence implicated that TIPP could take actions within cells. Interestingly, it was found that TIPP accumulated in the nucleus after entering cells, meriting further experiments to explore the mechanism of action. In addition, compared to the untreated control, pretreating cells with TIPP for 20 min displayed a significant anti-inflammatory effect on IgE-induced sensitization, suggesting that intracellular TIPP reached the effective concentration (24.57 ng/mL) within 20 min. These results laid a solid foundation for future research on the intracellular target and mechanism of TIPP.

Nevertheless, this study has some limitations. Due to rapid elimination after intravenous injection (data not shown), the metabolism of TIPP in mice could not be carried out using intravenous injection, and therefore, a PK study was carried out with subcutaneous administration. Due to the anti-inflammatory activity of TIPP, it may be further used in other inflammatory diseases. In order to extend the half-life, the dosage form and structure modification of TIPP are worth exploring in the future.

## 4. Materials and Methods

### 4.1. Compounds and Reagents

TIPP (AEWCP, purity ≥ 99%, molecular weight 604.68 Da), IS (S4: AEWSP, purity ≥ 99%, molecular weight 588.61 Da), and FITC-TIPP (5-FITC-(Ahx)-AEWCP (Figure 1c), purity ≥ 96%, molecular weight 1107.19 Da) were synthesized by QIANGYAO Biological Technology Company (Wuhan, China). ACN, FA, DCM, isopropanol (IPA), and MeOH, of LC-MS grade, were from Merck Darmstadt, Germany. DTT, L-AA, PMSF, monoclonal anti-dinitrophenyl antibody produced in mouse, IgE isotype, clone SPE-7 (anti-DNP-IgE), dinitrophenyl-human serum albumin (DNP-HSA), 1,4-piperazinebis (ethanesulfonic acid) (PIPES), p-nitrophenyl-N-acetyl-β-D-glucosaminide (p-NAG), 1,1′-dioctadecyl-3,3,3′,3′-tetramethylindocarbocyanine perchlorate (DiI) dye, and Hoechst 33342 were purchased from Sigma-Aldrich (St. Louis, MO, USA). Minimum Essential Medium (MEM) and fetal bovine serum (FBS) were obtained from Gibco (Waltham, MA, USA). TDW was obtained from Milli-Q PLUS PF water purifying system (Millipore, Bedford, MA, USA). Rat IL-4 and histamine ELISA detection kits were produced by Elabscience Biotechnology Company (Wuhan, China).

### 4.2. Instrumentation and LC-MS/MS Conditions

#### 4.2.1. Liquid Chromatography

LC running with a Thermo Hypersil Gold C18 column (2.1 mm × 100 mm, 5 μm) was performed on a Shimadzu chromatograph equipped with a Nexera X2 pump, DGU 20A3 degasser, and Shimadzu CTO-10AS column oven. A Shimadzu Nexera X2 autosampler was set at 4 °C, and the LC column was maintained at 40 °C. The mobile phase (0.1% FA-ACN and 0.1% FA-TDW) was running at a flow rate of 0.2 mL/min in a linear gradient mode (Table 1). The total run time was 5 min. The injection volume of all samples was 2 μL. To prevent the carry-over of samples, the needle was washed before and after each injection with ACN: TDW: IPA: MeOH mixture solution (25: 25: 25: 25% *v*/*v*).

#### 4.2.2. Mass Spectrometry

MS detection was performed using API 5500 Q trap mass spectrometer (Applied Biosystems, Waltham, MA, USA) equipped with a Turbo Ion Spray^TM^ ESI source, in a positive MRM mode. The optimized MS parameters for detecting TIPP and S4 (IS) are presented in Table 6.

### 4.3. Optimization of Sample Preparation

#### 4.3.1. Optimization of Protein Precipitation

TIPP stock solutions were separately diluted in TDW or murine plasma at a concentration of 1 μg/mL. Samples (10 μL) were subjected to protein precipitation either immediately or after 10 min incubation at room temperature (around 25 °C). A series of reagents were used for protein precipitation, including MeOH (300 μL) and 2% FA-ACN of different volumes (300, 140, 100, and 70 μL). After brief vortex, the samples were centrifuged at 15,000 rcf at 4 °C for 10 min, and the supernatants were collected for chromatographic analysis. In order to increase the signal intensity of highly water-soluble TIPP in organic solvents, a group of liquid–liquid extraction was carried out to separate the aqueous phase from the organic phase. After treating with 300 μL of 2% FA-ACN, the resulting supernatants (300 μL) were mixed with 500 μL of DCM for backwashing. The mixture was briefly vortexed and centrifuged at 15,000 rcf for 10 min, and the supernatant (the aqueous phase) was collected. After sample preparation, the processed samples were stored at −80 °C until analysis. Before sampling, the prepared sample was thawed, vortexed, and placed in the autosampler.

#### 4.3.2. Stabilizers Screening

To choose the optimal stabilizer, the performance of DTT (30 mg/mL), L-AA (10 mg/mL), and PMSF (50 μg/mL) was evaluated. Plasma (90 μL) was mixed with each stabilizer (20 μL), and TIPP (10 μL, 5 μg/mL). The samples were subjected to protein precipitation either immediately or after 10 min incubation at room temperature. For protein precipitation, the samples added with 10 μL of S4 (3 μg/mL) were mixed with 300 μL of 2% FA-ACN and processed as discussed in Section 4.3.1. The aqueous layer was collected for chromatographic analysis. The recovery of TIPP with each stabilizer was determined. Moreover, the stabilizing effect of DTT was further evaluated with different concentrations (2.5, 5, 10, 20, 25, and 40 mg/mL), following the above protocol.

### 4.4. Preparation of Calibration Solutions of TIPP and Quality Control (QC) Samples

TIPP stock solution of 1 mg/mL in 5% ACN-TDW was prepared and stored at −80 °C. TIPP standard solutions and QC samples were freshly prepared on the day of analysis. The stock solution of TIPP was serially diluted with TDW to prepare working solutions, which were subsequently added in blank murine plasma, RBL-2H3 cell lysates, cell culture supernatants, and the final cell rinse, respectively. The calibration quantification range and the LLOQ of TIPP in different samples, as well as concentration settings for LQC, MQC, and HQC samples are shown in Table 7.

### 4.5. Sample Preparation

#### 4.5.1. Preparation of Plasma Samples

Female BALB/c mice (6–8 weeks old) were purchased from Beijing Vital River Laboratory Animal Technology Co., Ltd. (Beijing, China). The animal experiment was carried out in accordance with Animal Experiment Guidelines of Shandong University, and the protocol was approved by the Animal Ethics Committee of Shandong University (Approval Number, 18001).

Whole blood was collected from inner canthus into a tube containing ethylenediaminetetraacetic acid (EDTA) and centrifuged at 5000 rcf for 3 min. Subsequently, 10 μL of plasma samples were diluted with 90 μL TDW. The sample processing process is then executed, and the diluted samples were mixed with 10 μL of S4 solution (3 μg/mL) and 300 μL of 2% FA-ACN in a tube containing 20 μL of DTT solution (25 mg/mL). After centrifugation at 15,000 rcf for 10 min, 300 μL supernatant was transferred to a new tube containing 500 μL of DCM. The mixture was vortexed and centrifuged at 15,000 rcf for 10 min, and the aqueous layer was loaded into the LC-MS/MS system. This entire process is shown in Figure 7a.

#### 4.5.2. Preparation of Cell Culture Supernatant, the Final Cell Rinse, and Cell Lysate Samples

RBL-2H3 cells were obtained from the Cell Resource Center of Shanghai Institutes for Biological Sciences (Shanghai, China) and cultured in complete MEM containing 15% heat-inactivated FBS in a 37 °C incubator (5% CO_2_ and 95% air). For the preparation of cell culture supernatant samples, as shown in Figure 7b, after centrifuging, the cell culture at 1000 rcf for 5 min, 10 μL of cell supernatant was diluted with 1990 μL of TDW, vortexed for 1 min, and 100 μL of the mixture was taken and prepared as that of plasma samples processing (4.5.1). To prepare final cell rinse samples, the cells were removed, the supernatant was rinsed with 1 mL of PBS five times, and 90 μL of the final cell rinse was collected and mixed with 10 μL of S4 solution (3 μg/mL) for chromatographic analysis directly. For the preparation of cell lysate samples, the cells (about 10^6^ cells) were lysed with 130 μL of RIPA buffer for 30 min on ice, and 100 μL of the cell lysate was collected for sample pre-processing following Section 4.5.1.

### 4.6. Methodology Validation

The LC-MS/MS method was validated in accordance with the Food and Drug Administration (FDA) guidelines for Bioanalytical Method Validation [22].

#### 4.6.1. Selectivity

The chromatograms of TIPP in murine plasma, cell lysates, cell culture supernatants, and the final cell rinse from six independent experiments were used to evaluate the selectivity. The intensity of the blank samples compared to their corresponding LLOQ and S4-spiked samples was required to be within ±20% and 5%, respectively.

#### 4.6.2. Linearity

The linearity was assessed by analyzing a series of standard samples. Murine plasma, cell lysates, cell culture supernatants, and the final cell rinse were mixed with TIPP stock standard solution to make final concentrations in the range of 20–5000 ng/mL, 1–200 ng/mL, 10–200 μg/mL, and 0.1–20 ng/mL, respectively (Table 7). A linear regression fitting was used; based on the analyte/IS peak area ratio versus the TIPP nominal concentration to the IS, the goodness of fit of the regression was calculated with Pearson’s determination coefficient *r*^2^ by the weighting factor 1/x^2^.

#### 4.6.3. Precision and Accuracy

HQC, MQC, LQC, and LLOQ samples (*n* = 6) in murine plasma, cell lysates, cell culture supernatants, and the final cell rinse were analyzed on the same day of preparation to determine the intra-assay precision, and the samples were analyzed on three consecutive days to determine the inter-assay precision. TIPP was quantified according to the calibrator determined on the same day, and it was compared with its nominal concentration to assess the accuracy of the quantitative methods. Precision was defined as the percent relative standard deviation (%RSD) with acceptance criteria of ±15% (except ±20% at LLOQ). Accuracy was defined as the percent bias (%Bias) with the same acceptance criteria as precision. %Bias was calculated according to the following Equation (1):(1)$$\%\mathrm{Bias}=\frac{\mathrm{observed}\;\mathrm{concentration}-\mathrm{nominal}\;\mathrm{concentration}}{\mathrm{nominal}\;\mathrm{concentration}}\times 100\%$$

#### 4.6.4. Matrix Effect and Recovery

Blank murine plasma, RBL-2H3 cell lysates, and cell culture supernatants from six independent experiments were mixed with TIPP solution at concentrations corresponding to LQC and HQC. The matrix effect on IS was performed in parallel samples. The absolute MF and IS normalized MF were calculated by Equations (2) and (3), respectively. The variability for IS normalized MF could be acceptable if the percentage of coefficient of variation (%CV) is within ±15%.
(2)$$\mathrm{MF}=\frac{\mathrm{Mean}\;\mathrm{peak}\;\mathrm{area}\;\mathrm{of}\;\mathrm{analyte}/\mathrm{IS}\;\mathrm{in}\;\mathrm{the}\;\mathrm{presence}\;\mathrm{of}\;\mathrm{matrix}\;\mathrm{components}}{\mathrm{Mean}\;\mathrm{peak}\;\mathrm{area}\;\mathrm{of}\;\mathrm{analyte}/\mathrm{IS}\;\mathrm{in}\;\mathrm{TWD}\;\mathrm{solvent}}$$
(3)$$\%\mathrm{IS}\;\mathrm{normalized}\;\mathrm{MF}=\frac{\mathrm{MF}\;\mathrm{of}\;\mathrm{analyte}}{\mathrm{MF}\;\mathrm{of}\;\mathrm{IS}}\times 100\%$$

The extraction recovery was assessed with LQC, MQC, HQC, and IS samples in murine plasma, RBL-2H3 cell lysates, and cell culture supernatants (*n* = 3), which was the ratio of the mean peak area of an analyte spiked before extraction to that of post-extraction.

#### 4.6.5. Stability

The stability of TIPP in plasma samples was investigated under different conditions. To evaluate autosampler stability, long-term stability, and short-term stability, up to three replicates of LQC and HQC plasma samples were evaluated when the samples were stored at room temperature for 4 h, at 4 °C for 24 h, and at −80 °C for 4 weeks, respectively. Freeze–thaw stability was also assessed in three cycles from −80 to 25 °C. Concentrations were determined using a TIPP calibrator of the respective validation day. Obtained results were compared to nominal concentrations to determine the stability.

### 4.7. PK Study in Mice

After 4 h fasting (*ad libitum* water intake), mice were subcutaneously injected with 50 mg/kg TIPP. At 0.5, 1, 2, 5, 10, 15, 20, and 40 min post administration (eight mice at each time points), blood samples were taken from an inner canthus into tubes containing Na_2_·EDTA and centrifuged at 5000 rcf for 10 min to obtain plasma. All samples were processed as described in Section 4.5.1 immediately. The scheme is shown in Figure 8a.

PK parameters were calculated as mean ± standard deviation (SD). The *C*_max_ and the time to reach *T*_max_ were calculated based on the highest concentration and the corresponding time point, and *t*_1/2_ was calculated from *C*_0_ to *C*_40_. The area under the curve during the 40 min window (*AUC*_0→40_) and the extrapolated value (*AUC*_0→∞_) were calculated using the trapezoidal rule, and the elimination rate and concentration at 40 min were extrapolated from 40 min to infinity. The CL/F and *V*_d_/F were calculated using the first-order one-compartment model.

### 4.8. Uptake Study of TIPP by RBL-2H3 Cells

#### 4.8.1. Determination of TIPP Concentration in RBL-2H3 Cell Lysate, Supernatant, and Final Cell Rinse Samples

RBL-2H3 cells were seeded in a 35 mm diameter dish at 8 × 10^5^ cells and treated with 200 μg/mL TIPP for different durations (5 min, 10 min, 20 min, 30 min, 40 min, 1 h, 3 h, 6 h, 12 h, and 24 h). Then, the cells were subjected to the pre-processing procedures to prepare cell lysates, cell culture supernatants and the final cell rinse samples (Section 4.5.2). In addition, to study TIPP intracellular degradation, the cells were incubated with TIPP for 40 min, which was followed by incubation with TIPP-free medium for 10, 20, 30, 40, and 60 min, respectively. The cell lysates were then harvested at the above time points and pre-processed as presented in Section 4.5.2. The flow charts are shown in Figure 8b.

#### 4.8.2. Visualization of Cellular Localization of Labeled TIPP

RBL-2H3 cells (4 × 10^4^) were seeded in a 35 mm confocal dish and treated with 367 μg/mL FITC-TIPP for 0, 5, 10, 20, 30, 40, and 60 min, respectively. Then, the cells were fixed with 4% paraformaldehyde solution for 10 min and permeabilized with 0.1% Triton-100 for 5 min, which was followed by nuclear staining in a dark place with Hoechst 33342 for 5 min. Finally, the cells were washed, stained with DiI dye solution for 10 min, rinsed with PBS three times, and analyzed with a confocal microscope.

### 4.9. The Effects of TIPP on Timing Inhibiting Inflammatory Factor In Vitro

#### 4.9.1. Cell Sensitization and Stimulation for Degranulation Assay

RBL-2H3 cells were inoculated in a 24-well plate at 8 × 10^4^ cells per well, and challenged with 0.2 μg/mL anti-DNP-IgE overnight to induce sensitization. After rinsing with PBS, the cells were treated with 200 μg/mL TIPP prepared with PIPES buffer (119 mM NaCl, 5 mM KCl, 25 mM PIPES, 5.6 mM glucose, 1 mM CaCl_2_, 0.4 mM MgCl_2_, and 0.1% BSA, pH 7.2) for 5, 10, 20, 30, 40, and 60 min, which was followed by incubation with 1 μg/mL DNP-HSA for 15 min. The cell culture supernatants were collected for subsequent assays.

#### 4.9.2. Determination of β-Hexosaminidase Activity

The cell culture supernatant (25 μL) from the previous study (Section 4.9.1) was transferred to a new tube containing 25 μL of p-NAG solution (10 mmol/L, in 0.1 mmol/L sodium citrate buffer, pH 4.5) and incubated at 37 °C for 1 h. The reaction was terminated with 0.1 mmol/L sodium carbonate buffer (pH 10.0), and the absorbance was measured at 405 nm [5].

#### 4.9.3. Quantification of Histamine and IL-4

Concentrations of histamine and IL-4 in cell culture supernatants collected in Section 4.9.1 were measured by a histamine ELISA detection kit according to the manufacturer’s instruction.

### 4.10. Analysis and Statistics

Analyst^®^ 1.5 software (AB SCIEX) was used for HPLC-MS/MS data processing. A non-compartmental model was employed to fit the PK of TIPP in mice plasma using DAS 2.0 software. GraphPad Prism was employed for statistical analysis and one phase exponential decay model analysis in RBL-2H3 cells. Results were compared using *t* test or one-way analysis of variance, which was followed by Dunnett’s multiple comparisons test, with *p* < 0.05 considered significant.

## 5. Conclusions

In summary, a rapid LC-MS/MS-based method with high selectivity and sensitivity for the quantitative analysis of TIPP in murine plasma and RBL-2H3 cells has been successfully developed and validated. The proposed LC-MS/MS method was validated according to FDA guidelines, and it showed acceptable selectivity, accuracy, and precision, with a negligible matrix effect. In addition, the validated method was successfully adopted to evaluate the in vivo metabolism and in vitro cell entry of TIPP in biological samples.

## Figures and Tables

**Figure 1 molecules-27-04256-f001:**
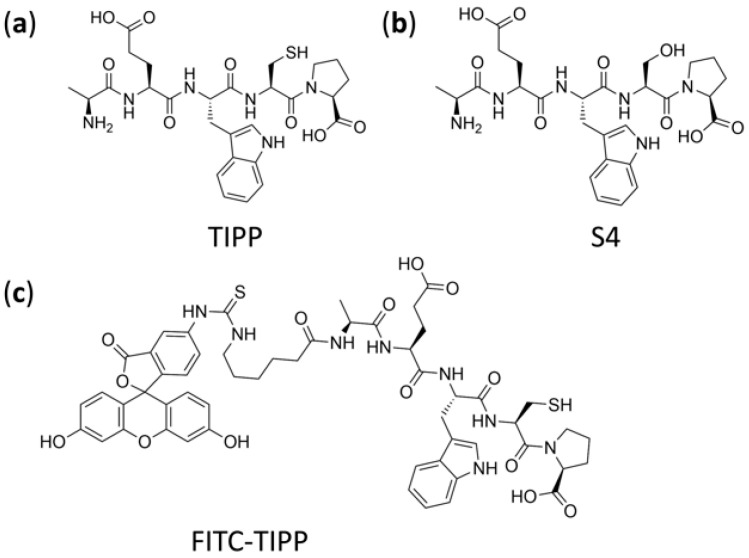
Chemical structure of TIPP (**a**), internal standard S4 (**b**), and FITC-TIPP (**c**).

**Figure 2 molecules-27-04256-f002:**
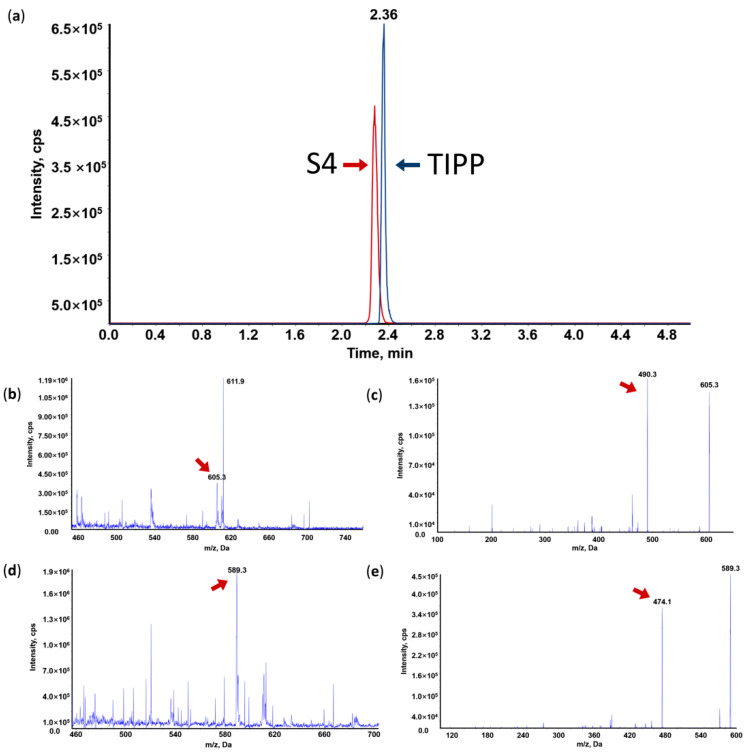
Representative MRM chromatogram of TIPP and internal standard (IS) (**a**); Mass spectrum (MS1), tandem mass spectrum (MS2) and representative MRM chromatogram of TIPP and S4. (**b**) MS1 of TIPP; (**c**) MS2 of TIPP; (**d**) MS1 of IS; (**e**) MS2 of IS.

**Figure 3 molecules-27-04256-f003:**
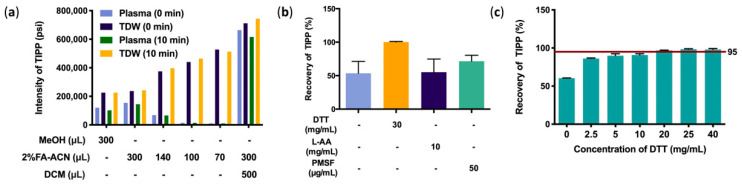
Optimization of sample preparation methods. Optimization of protein precipitation (**a**), effects of stabilizers on recovery of TIPP (**b**), and effects of DTT on recovery of TIPP (*n* = 3) (**c**).

**Figure 4 molecules-27-04256-f004:**
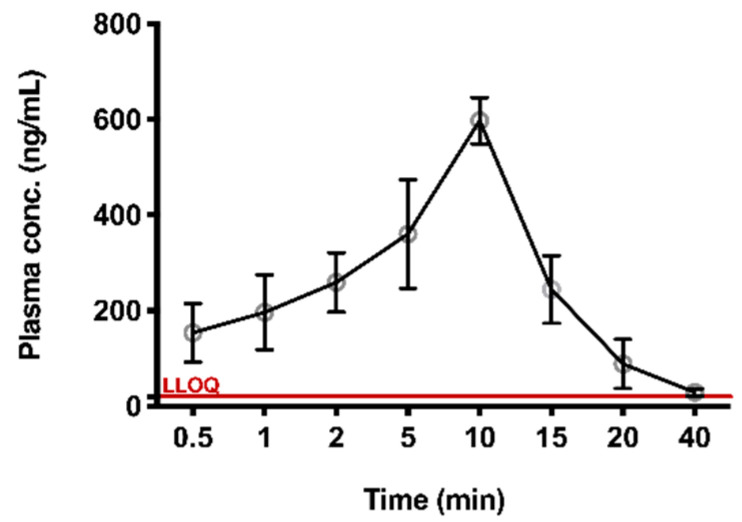
The plasma concentration versus time profile for TIPP. Plasma concentration-time profile after 50 mg/kg subcutaneous administration of TIPP (mean ± SD, *n* = 8 at each time point).

**Figure 5 molecules-27-04256-f005:**
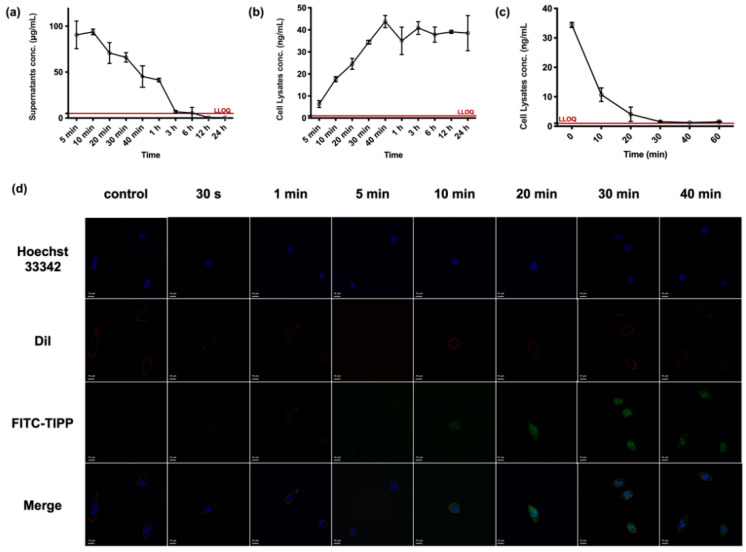
The uptake of TIPP in RBL-2H3 cells. The concentration versus time profiles of TIPP in RBL-2H3 cells related matrix samples at a dose of 200 μg/mL (*n* = 3). The concentration–time curve of TIPP in RBL-2H3 cell culture supernatants (**a**) and lysates (**b**). After substituting with a TIPP-free medium, the decreased TIPP concentration in cell lysates (**c**). Confocal microscopic pictures of the uptake of FITC-TIPP by RBL-2H3 cells over time (**d**). The green, blue, red, and merge were signals of FITC-TIPP, nucleus, membrane, and the merging, respectively. Magnification: 200×.

**Figure 6 molecules-27-04256-f006:**
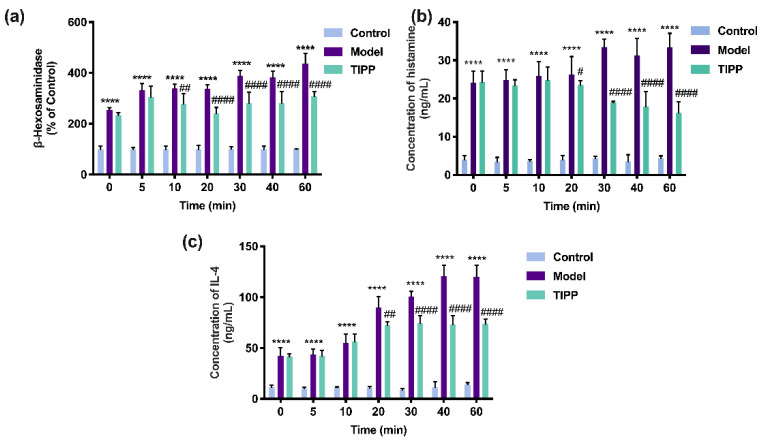
Effects of TIPP on IgE-mediated degranulation in RBL-2H3 cells. The effects of stimulated time of TIPP on β-hexosaminidase release (**a**), in which the cells stimulated with IgE-antigen complex without TIPP were used as a control of 100%. Effects of TIPP on DNP–HSA-induced histamine (**b**) and interleukin-4 release (**c**). Results are expressed as mean ± SD (*n* = 3), **** *p* < 0.0001 vs. control, ^#^
*p* < 0.05 vs. model, ^##^ *p* < 0.01 vs. model, ^####^
*p* < 0.0001 vs. model.

**Figure 7 molecules-27-04256-f007:**
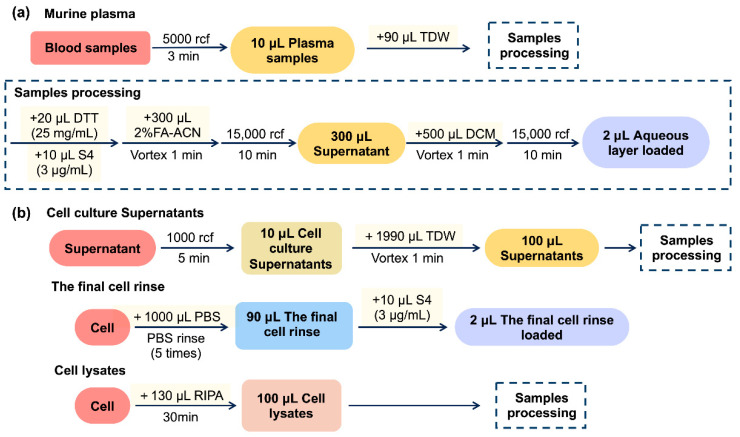
Scheme of sample preparation of TIPP in mice (**a**) and RBL-2H3 cells (**b**).

**Figure 8 molecules-27-04256-f008:**
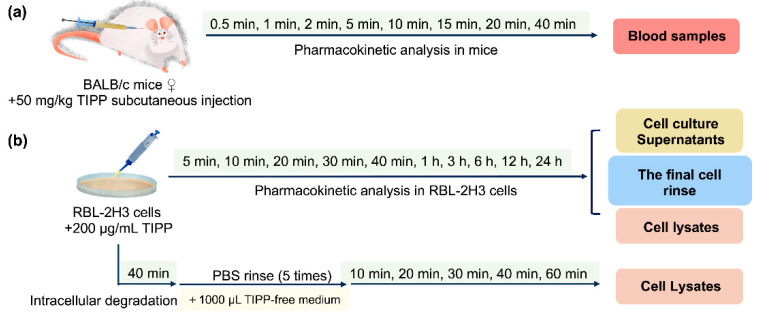
Flow charts of pharmacokinetic analysis of TIPP in mice (**a**) and RBL-2H3 cells (**b**).

**Table 1 molecules-27-04256-t001:** Gradient elution for quantification of TIPP. Mobile phase A consisted of TDW and mobile phase B of ACN, both containing 0.1% (*v*/*v*) formic acid.

Time, min	Mobile Phase A, %	Mobile Phase B, %
0	90	10
0.5	90	10
2	10	90
2.5	10	90
3	90	10
5	90	10

**Table 2 molecules-27-04256-t002:** Intra-assay and inter-assay precision and accuracy for TIPP-spiked mouse plasma, RBL-2H3 cell culture supernatants, lysates, and the final cell rinse (*n* = 6).

Samples		Accuracy		Precision	
		%Bias	%Bias	%RSD	%RSD
		Intra-Assay	Inter-Assay	Intra-Assay	Inter-Assay
Mouse plasma	LLOQ	0.97	0.22	4.38	3.24
	LQC	1.07	1.71	7.04	5.38
	MQC	−1.42	−1.19	4.53	6.63
	HQC	−0.19	−1.06	4.21	4.47
Cell culture supernatants	LLOQ	−7.43	−1.91	3.97	4.41
	LQC	−3.33	−4.00	3.52	2.79
	MQC	−3.81	−2.51	3.21	2.75
	HQC	2.04	−0.19	2.77	3.3
Cell lysates	LLOQ	1.25	2.03	5.65	5.17
	LQC	−6.17	−5.28	1.56	2.87
	MQC	1.88	1.38	2.5	2.35
	HQC	3.02	4.9	2.86	2.28
The final cell rinse	LLOQ	−8.22	2.07	10.62	7.91
	LQC	−3.38	−2.18	6.15	5.05
	MQC	−0.35	−2.12	6.27	4.38
	HQC	1.35	−2.22	4.86	4.94

**Table 3 molecules-27-04256-t003:** Extraction recovery and IS-normalized matrix factor for TIPP-spiked samples.

	% Extraction Recovery (Mean ± SD, *n* = 3)			% IS-Normalized Matrix Factor (Mean ± SD, *n* = 6)		
	Mouse Plasma	Cell Culture Supernatants	Cell Lysates	Mouse Plasma	Cell Culture Supernatants	Cell Lysates
LQC	41.3 ± 6.5	94.1 ± 6.7	59.8 ± 5.8	104.7 ± 13.7	113.7 ± 3.5	109.4 ± 14.8
MQC	42.1 ± 1.8	93.5 ± 6.5	57.4 ± 4.9	102.6 ± 2.2	108.2 ± 2.3	99.5 ± 9.6
HQC	41.1 ± 1.6	86.9 ± 3.7	58.7 ± 4.6	100.3 ± 5.8	109.7 ± 5.9	97.9 ± 4.4
IS	42.6 ± 4.6	80.2 ± 5.1	68.2 ± 4.7	-	-	-

**Table 4 molecules-27-04256-t004:** Stability results of TIPP in murine plasma under different conditions (mean ± SD, *n* = 3).

Conditions	LQC	HQC
4 h at room temperature	96.00 ± 0.01	98.02 ± 0.04
24 h at 4 °C	103.64 ± 2.00	99.35 ± 0.84
4 weeks at −80 °C	90.70 ± 1.61	90.70 ± 0.12
Three freeze-thaw cycles (at −80–25 °C)	94.23 ± 0.02	106.00 ± 0.16

**Table 5 molecules-27-04256-t005:** TIPP pharmacokinetic parameters in mice after 50 mg/kg subcutaneous administration (*n* = 8).

Parameter	Mean ± SD
*AUC*_(0–t)_, μg/L×min	7982.05 ± 1488.44
*AUC*_(0–∞)_, μg/L×min	8182.81 ± 1529.9
*MRT*_(0–t)_, min	11.32 ± 1.29
*MRT*_(0–∞)_, min	12.53 ± 1.46
*t*_1/2_, min	5.99 ± 1.82
*T*_max_, min	10.00 ± 0.00
CL/F, L/min/kg	6.34 ± 1.45
*V*d/F, L/kg	54.39 ± 20.95
*C*_max_, μg/L	623.00 ± 67.13

**Table 6 molecules-27-04256-t006:** Optimized mass-dependent parameters of TIPP and S4.

Source Parameters	Value	
Source temperature, °C	550	
Curtain gas, psi	30	
Collision gas, psi	7	
Ion spray voltage, eV	5500	
Nebulizer gas (gas1), psi	25	
Turbo Ion gas (gas2), psi	12	
Polarity	Positive	
**Compound Parameters**	**TIPP**	**S4 (IS)**
Precursor ion, *m*/*z*	605.3	589.3
Product ion, *m*/*z*	490.2	474.2
Declustering potential, eV	130	130
Collision energy, eV	30	25
Collision cell exit potential, eV	10	5
Entrance potential, eV	8	8

**Table 7 molecules-27-04256-t007:** Concentrations of TIPP in QC samples and calibration curve in mouse plasma samples, cell culture supernatants, cell lysates, and the final cell rinse PBS.

Samples	LLOQ	LQC	MQC	HQC	Calibration Curve						
Mouse plasma, ng/mL	20	50	2000	4000	20	100	250	500	1250	2500	5000
Culture supernatants, μg/mL	5	15	90	180	5	10	20	50	100	150	200
Cell lysates, ng/mL	1	4	40	160	1	5	10	25	50	100	200
The final cell rinse PBS, ng/mL	0.1	0.4	4	16	0.1	0.5	1	2.5	5	10	20

## Data Availability

Not applicable.

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
