# Peer review of "The Pharmacokinetics in Mice and Cell Uptake of Thymus Immunosuppressive Pentapeptide Using LC-MS/MS Analysis"

_molecules, 2022, doi:10.3390/molecules27134256_

Round 1

Reviewer 1 Report

Chen and coworkers performed nice work by developing an effective LC/MS method and applying it to examine the pharmacokinetics of an immunosuppressive pentapeptide in subcutaneous mice and in cultured cells. The experiments are well designed and contain meaningful controls. The data are solid, presented clearly, and support the conclusions. The results on pharmacokinetics are very interesting and important.

Minor critics:

The image resolution of figures 2 and S1 was poor and should be improved.

Figure 2 legend: better explain what is exactly the “IS”.

Page 4, line 108: change ‘protonate’ to ‘protonation’

Page 9, line 220, change the word aggregate to either enrichment or concentration.

Figure 6: what is the difference between control and model?

Reviewer 2 Report

The authors describe a novel LC-MS/MS method for quantification of the pentapeptide TIPP that was validated followinf FDA guidelines and successfully applied to a PK analysis of TIPP in mice. The paper is very well written, the method development experiments  are well described and validation protocol was complete. The manuscript can be considered acceptable in its present form with any revision required.

Author Response

Dear Reviewer:

Thank you very much for your kind comments on our manuscript. There is no doubt that these are valuable and very helpful for revising and improving our manuscript.

 If you have any questions, please contact us without hesitation.

Yours sincerely,

Professor Fengshan Wang

This manuscript is a resubmission of an earlier submission. The following is a list of the peer review reports and author responses from that submission.